# Evaluation of Mechanical, Physical, and Morphological Properties of Epoxy Composites Reinforced with Different Date Palm Fillers

**DOI:** 10.3390/ma12132145

**Published:** 2019-07-03

**Authors:** Basheer A. Alshammari, Naheed Saba, Majed D. Alotaibi, Mohammed F. Alotibi, Mohammad Jawaid, Othman Y. Alothman

**Affiliations:** 1Materials Science Institute, King Abdulaziz City for Science and Technology (KACST), Riyadh 11442, Saudi Arabia; 2Department of Bio Composite Technology, Institute of Tropical Forestry and Forest Products, Universiti Putra Malaysia, 43400 UPM Serdang, Selangor, Malaysia; 3Life Science and Environmental Research Institute, King Abdulaziz City for Science and Technology (KACST), Riyadh 11442, Saudi Arabia; 4Department of Chemical Engineering, College of Engineering, King Saud University, Riyadh 11421, Saudi Arabia

**Keywords:** date palm tree, leaf sheath fiber, epoxy, water absorption test, mechanical strength, morphological properties

## Abstract

The present study deals with the fabrication of epoxy composites reinforced with 50 wt% of date palm leaf sheath (G), palm tree trunk (L), fruit bunch stalk (AA), and leaf stalk (A) as filler by the hand lay-up technique. The developed composites were characterized and compared in terms of mechanical, physical and morphological properties. Mechanical tests revealed that the addition of AA improves tensile (20.60–40.12 MPa), impact strength (45.71–99.45 J/m), flexural strength (32.11–110.16 MPa) and density (1.13–1.90 g/cm^3^). The water absorption and thickness swelling values observed in this study were higher for AA/epoxy composite, revealing its higher cellulosic content, compared to the other composite materials. The examination of fiber pull-out, matrix cracks, and fiber dislocations in the microstructure and fractured surface morphology of the developed materials confirmed the trends for mechanical properties. Overall, from results analysis it can be concluded that reinforcing epoxy matrix with AA filler effectively improves the properties of the developed composite materials. Thus, date palm fruit bunch stalk filler might be considered as a sustainable and green promising reinforcing material similarly to other natural fibers and can be used for diverse commercial, structural, and nonstructural applications requiring high mechanical resistance.

## 1. Introduction 

A composite is a material formed of two or more physical components, offering much better properties than those of the single components, i.e., the matrix and the filler. Composites are usually manufactured from various types of matrices, such as polymers, metals, and ceramics. Among them, polymers have advantageous properties, such as chemical resistance, low density, good wettability, and easy molding into desired and engineered shapes. Therefore, polymers are used as matrices to a greater extent, compared to metals and ceramics. However, polymers have lower strengths and moduli [1,2]. To overcome this problem, the introduction of additives and natural fillers (of plant, animal, and mineral origin) is the most familiar technique to improve the mechanical properties. Plant fibers are mainly sourced from the leaves, stems, and seeds of a variety of plants, such as date palm tree, bamboo, jute, flax, hemp, sisal, kenaf, coir, kapok, and banana. Their major components are cellulose, hemicellulose and lignin [3,4].

The introduction of additives and natural fillers to reinforce composites has a number of advantages, such as good strength, recyclability, renewability, biodegradability of the developed materials, as well as lower energy consumption for their fabrication and local availability of the raw materials. However, the high-water absorption and low mechanical properties of natural fibers limit some of their applications [5,6,7]. Until now, several researchers have reviewed the efficiency of using natural fibers as sustainable reinforcing materials for both thermoplastic and thermoset polymers intended for construction and automotive applications [8,9,10]. Researchers have investigated the use of natural fiber to improve the mechanical properties of polymer composites [3]. Many findings also reported the tensile properties of natural fiber reinforced polymer composites, considering their potential industrial applications [4,11]. 

The annual worldwide production of the date palm tree [family Palmae (Arecaceae)] is about 42%, i.e., 20% and 10% higher than the production of coir and sisal/hemp, respectively [12,13]. The tree is widely grown in the tropical and subtropical regions, in Saudi Arabia and Asia, being part of the daily life of the people inhabiting the Arabian Peninsula for the last 7000 years [12,13]. A structural image of the date palm tree and its different components is illustrated in Figure 1a–c [14,15,16]. 

Remarkably, date palm trees produce a great amount of waste materials. Several studies have analyzed various date palm residues and their suitability to be utilized as filler in polymer composites [17,18]. Generally, virgin and recycled thermoplastics are widely used as matrices for date palm fiber (DPF). For instance, a researcher group examined the mechanical properties of DPF reinforced polyethylene (PE) composites [19]. Others investigated the sorption, mechanical and adhesive properties of date palm wood powder/low-density polyethylene composites (LDPE) [20]. In addition, Almaadeed et al. [14] reported an improvement in the mechanical properties of recycled LDPE composites filled with date palm wood powder. Researchers have also focused on the natural and artificial weathering degradation of DPF/polypropylene composites [21] and in polyester matrix [22]. In another study, the mechanical and thermal properties of DP leaf fibers/recycled polyethylene terephthalate (PET) composites were reported [23]. Recently, the influence of DPF on the mechanical, physical and thermal properties of LDPE composites was described [24]. Some of the recently reported works on the date palm material reinforced polymer composites are tabulated in Table 1.

From Table 1 above, it is evident that the potential of date palm fiber/filler as reinforcement for polymeric composites is well known [42]. However, very few studies have been performed on DPF reinforced thermosetting polymers. Recently, the mechanical and thermal properties of DPF/epoxy composites have been reported in the literature [13]. The researchers concluded that 50 wt% fiber loading is optimum to enhance the mechanical and thermal properties of epoxy composites. 

Considering that date palm trees are widely available in Saudi Arabia, the present study focuses on the potential of fibers obtained from various date palm tree parts to be used as an additive in the manufacture of polymer composites. The objective of the current study is to develop sustainable green products that can be used in construction and building applications. Thus, fibers from the leaf sheath, leaf stalk, fruit bunch stalk and trunk of date palm tree (*Phoenix dactylifera* L.) were used as filler reinforcement for an epoxy matrix at 50 wt% loading. The fabricated composites were characterized, compared, and the data were interpreted to determine the most promising date palm material to reinforce epoxy composites. The findings of the study might reveal a more sustainable approach to address the problem of date palm biomass deposition in the Saudi Arabian region by using these residues as sustainable and promising filler in the manufacture of different polymers. In addition to this, the success of this study will result in more environmentally friendly and sustainable products that could help to replace some hydrocarbon usage. 

## 2. Experimental Procedure

### 2.1. Materials 

In this study, epoxy and Jointmine 905-3S were obtained from Tazdiq Engineering Sdn. Bhd., Kajang, Malaysia, while different DPF were procured from Riyadh City, Saudi Arabia. The date palm tree parts from where the fibers are extracted and used in this study are displayed in Figure 2.

The chemical composition of the extracted DPF was investigated in accordance with TAPPI standards [25] and the results were tabulated in Table 2.

### 2.2. Fabrication of Composites

In this study, four varieties of ground DPFs (0.8–1 mm) were used as filler to fabricate DPF/epoxy composites by the hand lay-up method using a steel mold 150 mm × 150 mm × 3 mm, while maintaining a total fiber loading of 50 wt%. The mold was transferred into a hot press, heated to a temperature of 110 °C for 10 min, and then transferred into a cold press for 5 min before demolding. 

## 3. Characterizations

### 3.1. Flexural Test

The three-point bending flexural tests were carried out on INSTRON Universal Testing Machine (Instron 5567, Shakopee, MN, USA) as per ASTM D-790 (2010) standard with a 5-kN capacity [43,44], at the relative humidity of 50 ± 5% and temperature of 23 ± 1 °C on each of the six replicate rectangular composite specimens having dimension of the order of 120 mm × 20 mm × 3 mm with a crosshead speed of 2 mm/min and at the gage length of 50 mm.

### 3.2. Tensile Test

Tensile tests were performed through Universal Testing Machine (Instron 5567, Shakopee, MN, USA) having maximum load cell of 5 kN capacity, as per ASTM 3039 (2014) standard [45], using five replicate specimens, with dimensions of 120 mm × 20 mm × 3 mm, at a crosshead speed of 2 mm/min and gauge length of 50 mm. The support span was 16 times the specimen depth and the testing speed was calculated using Equation (1). Prior to test the samples were placed in a conditioning chamber for one day at 23 ± 3 °C and relative humidity of 50 ± 10%. Data were analyzed and the average results were interpreted.
R = 0.01L^2^/6d(1)
where R is the rate of crosshead motion (mm/min), L is the support span (mm), and d is the depth of beam (mm).

### 3.3. Impact Test 

Prior to the Izod impact test (Type A), V-shaped notches were made by using NOTCHVIS (CEAST) for the five replicate specimens of each sample having the dimensions of 70 mm × 15 mm × 3 mm as per the ASTM D256 (2010) standard [46,47] through Gotech GT-7045-MD (Taichung City, Taiwan). The notch angle had a 45° radius and the notch depth was 2.5 mm. Suitable pendulum hammers imparting a 3.56 kg impact were mounted with the maximum impact energy of 5.0 J and the machine was calibrated for energy and accurate determination of the exact amount of impact energy (J/m) involved in the tests. The energy needed to break the composite specimen, its toughness, and average impact energy was then analyzed and reported. 

### 3.4. Water Absorption Test

Water absorption tests were carried out for each specimen, with dimensions of 20 mm × 20 mm × 5 mm, as per ASTM D 570-98 (2010) [48,49,50,51]. The initial weight of the test specimen (W_d_) was measured and recorded before immersion into distilled water. Then, the weight of the test specimen (W_n_) was measured again and recorded every 24 h for a week. The water absorption values of the composites were calculated using Equation (2):Water absorption (%) = (W_n_ − W_d_/W_d_) × 100(2)
where W_n_ is the weight of the composite samples after immersion and W_d_ is the weight of the composite samples before immersion into distilled water.

### 3.5. Thickness Swelling Test

Thickness swelling of each composite specimen, with dimensions of 20 mm × 20 mm × 5 mm, was assessed as per ASTM D 570-98 (2010) [52,53]. The initial thickness of the test specimen was measured and recorded before it was immersed in distilled water. After immersion, the specimen thickness was measured and recorded every 24 h for a week. Thickness swelling of the samples was calculated using Equation (3):Thickness Swelling (%) = (T_1_ − T_0_/T_0_) × 100(3)
where T_1_ is the thickness after soaking and T_0_ is the thickness before soaking.

### 3.6. Scanning Electron Microscopy (SEM)

The fractured surface morphology of the samples was analyzed using an EM-30AX scanning electron microscope (SEM, COXEM, Daejeon, Korea), with an acceleration voltage of 20 kV. Prior to the scanning, the samples were coated with a thin layer of gold.

## 4. Results and Discussion 

### 4.1. Flexural Properties

Figure 3 illustrates the effect of adding 50 wt% of each type of DPF to the epoxy matrix on their flexural properties. Figure 3 clearly reveals that the incorporation of DPF remarkably improves the flexural properties, compared with those of pure epoxy. The flexural properties of the composites containing AA filler are attributed to the high toughness and stiffness of this type of DPF, compared to the corresponding properties of pure epoxy composites.

The flexural modulus of the composites increased by ~60% initially and remained almost unchanged for all the formulations. The AA/epoxy composites exhibit better flexural properties, compared to rest of the composites, which can be explained by the perfect wetting of DPF, better dispersion, and good interfacial bonding between the filler and the epoxy matrix, which led to the formation of fewer voids and less fiber breakage. Other research works suggested that composites with lower lignin content possess better flexural properties, owing to better adherence between the fibers and the polymer matrix [54], thus providing an effective stress transfer between the reinforcement fibers and the matrix [55]. This is also in agreement with the findings of another study, which concluded that composites with lower lignin content exhibit better flexural strength [19]. Research has established that the flexural strength of polymer composites is affected by the properties of the fiber and the matrix, the interfacial interaction between them and the homogeneity of the composites [56]. In contrast, the flexural modulus depends on the filler, support span and environmental conditions. 

The flexural strengths of all DPF/epoxy composites show higher values, compared to their tensile strength (Figure 4), which might be due to the orientation of DPF in the outer layer of the composites. Similar results have been also observed for date palm leaf fiber/recycled PET composites [23]. In another study, the flexural properties of DPF/epoxy composites were analyzed and it was reported that the reinforcement using DPF remarkably enhanced the flexural properties of epoxy composites [13]. 

### 4.2. Tensile Properties

Figure 4 presents the comparative tensile strength and modulus of the DPF/epoxy composites under examination. It is clear from Figure 4 that the tensile properties present a similar trend to that of the flexural properties (Figure 3). It is obvious that the tensile properties of the composites exhibit higher values than that of the pure epoxy matrix. The highest tensile strength value is observed for the composites containing AA filler. The tensile strength of AA/epoxy recorded an increment from 20.60 to 40.12 MPa, while its tensile modulus increased from 0.57 to 2.88 GPa, compared to the rest.

The superior tensile properties of AA/epoxy composites can be ascribed to the properties of DPF, containing higher cellulose and hemicellulose content, besides the better interfacial interaction and dispersion of the filler in the polymer [19]. Comparable opinions were reported in the literature revealing that *Areca* fruit husk fibers, which had higher cellulose content, displayed better tensile strength [57]. In another study, cassava bagasse and sugar palm fiber were used as reinforcement in cassava starch and fructose (plasticizer) composites prepared by the casting technique [58]. The authors also declared that cellulose is the prime structural component providing strength to the stem walls and plant fiber.

However, the tensile modulus is less sensitive to these parameters, compared with tensile strength. Tensile modulus shows a logical trend for the composites containing low stiffness polymer and high stiffness natural filler [56]. An earlier study reported that, as the pineapple leaf fibers possess the highest cellulose content (70–82%) among most of the natural fibers, it governs their superior tensile modulus and tensile strength [59]. Another study on treated Ensete stem fiber reinforced polyester composites demonstrated their high cellulosic content, as well as better interfacial interaction, related to increased fiber surface roughness [60]. Other researchers investigated the tensile properties, impact strength and flexural properties of polymer composites reinforced with DPF/epoxy [13], DPF/PE [20], DPF/recycled PE [14], and DPF/PET [23], revealing a considerable improvement, particularly, in the moduli of the composite materials due to the addition of DPF. They also revealed that the mechanical properties get deteriorated as the DPF loading was increased beyond a certain level, as the composites became much more brittle.

### 4.3. Impact Strength

Figure 5 illustrates the effect of DPF addition as filler to the epoxy matrix on the impact strength of the materials. From Figure 5, it is evident that the addition of DPF leads to higher impact properties, compared with those of the pure epoxy composites, indicating the positive effects of the reinforcement. The observed lower impact strength of pure epoxy justifies its brittle nature [61]. The impact strength properties of the DPF/epoxy composites were found to be somewhat higher, owing to the stiffer nature of the date palm filler compared with pure epoxy. This result is in agreement with the findings of other studies, which reported that the reinforcement of an epoxy matrix with oil palm filler enhanced the impact strength of the obtained composites [62].

Researchers also reported that the impact strength of composites is greatly influenced by the toughness properties of DPF, the interfacial adhesion, as well as the frictional work that is needed to pull out the fiber from the matrix [24,63]. The findings are in agreement with those described in another study regarding the analysis of mechanical testing results for recycled polystyrene/wood flour composites, pointing out that interfacial adhesion and enlarged contact surface area have a profound effect on the flexural and impact strength of the materials [64]. Figure 5 also illustrates the energy absorption (%) along with impact strength of composites. Energy absorption is the area under a stress–strain curve, hence, it depends highly on the tensile strength of a material and it is regarded as a different means to evaluate toughness [65]. Interestingly, energy absorption mainly occurs during deformation and fracture processes [66]. Figure 5 clearly revealed that, besides the increase in impact strength by the addition of DPF to the epoxy, a remarkable decrease in energy absorption during the izod impact test was also observed for all reinforced composites. However, it is more pronounced for AA/epoxy with respect to other composites. Thus, lower energy absorption and higher damage resistance tendency are realized for AA/epoxy relative to the rest of the composites, are more likely associated with the amount of internal damage during impact load. All this governed that the AA/epoxy composites showed the highest content of cellulose in their chemical composition, which contributed to their higher mechanical strength and lesser internal damage during stress.

### 4.4. Water Absorption

Figure 6 shows the percentage water absorption of DPF/epoxy composites. Figure 6 reveals that the incorporation of DPF in the epoxy matrix increases their water absorption values, with increased soaking time, as the samples reach saturation after approximately six days. This behavior is explained by the hydrophilic nature of DPF, which is due to the presence of polar groups that create strong hydrogen bonding between the water molecules and the cellulose, as illustrated in Figure 7 [24].

This trend is typical of most natural fiber-reinforced polymer composites, revealing that DPF/epoxy composites have the tendency to absorb more moisture compared to the pure epoxy composites. Such behavior could be explained by the fact that water absorption is generally affected by the existence of lumens, pores, gaps, holes, voids, flaws, poor interfacial adhesion, and microcracks at the interface between the DPF and the epoxy matrix. Strong adhesion implies fewer locations in the composites that could store water molecules and therefore results in reduced water absorption. In contrast, higher water absorption leads to the formation of micro cracks, which finally result in the creation of voids and free spaces within the composites because of the swelling fibers [61]. The developed voids/free spaces contribute to a potential fracture mechanism in these composites.

### 4.5. Thickness Swelling

Figure 8 shows the percentage thickness swelling of the composite samples under examination. It is clear that the thickness swelling of the composites increases with water absorption and, thus, exhibits similar results, indicating a correlation between thickness swelling and weight gain due to water absorption. The observed trend is in agreement with that noted in previous findings with regard to sugar palm reinforced epoxy composites [67]. 

From Figure 8, it can be also observed that AA/epoxy shows higher thickness swelling behavior, owing to its higher cellulosic content, as in general the chemical composition (cellulose content) of fibers determines the physical and mechanical properties. These statements are also in line with those of other researchers [58,68]. Noticeably, research has established that thickness swelling minimizes interfacial adhesion [33,69], which governs the mechanical properties. Actually, the water absorption and thickness swelling of natural fiber reinforced polymer composites are some of their main drawbacks, since the presence of moisture results in the degradation of the fiber matrix interface or in delamination, fiber debonding and, finally, the formation of microvoids and interfacial cracks induced by moisture which, in turn, affect their mechanical properties. Complementary statements have been reported and reviewed by other researchers [70,71].

### 4.6. Density

Figure 9 displays the density of epoxy and DPF/composites. It is clear from the plot that reinforcing the epoxy matrix with DPF results in remarkable changes in their density. The density of pure epoxy composites increases from 1.13 g/cm^3^ to 1.9 g/cm^3^ upon the introduction of the DPF filler. However, the difference among the densities of the DPF/epoxy composites is not much higher. Among the DPF reinforced materials, the AA composites possess higher densities compared to the others, which can be explained by both the chemical and physical properties of palm tree fibers, including the bulk density of each type of filler [17,19]. The increase in the density of the composites further leads to an improvement in their hardness and toughness, which govern the suitability of these materials for light weight applications. This result is in line with the findings of a previous investigation [7], which reported that the incorporation of kenaf and oil palm nano filler improves the density of an epoxy based material and consequently its hardness. Another study evaluated the influence of DPF reinforcement in LDPE on the physical and mechanical properties of the obtained materials. The authors declared that the density of the composites increased with an increasing amount of fiber loading up to ~60 wt%, followed by a reduction in the values of density [23].

### 4.7. Morphological Analysis

Figure 10 and Figure 11 display the SEM images of fractured surfaces of pure epoxy and DPF/epoxy composites, respectively. SEM clearly reveals the surface morphology and microstructure of the composites, indicating the factors that determine earlier rupture during mechanical testing. Figure 10 presents a micrograph of epoxy composites, disclosing their brittle plastic nature, with a glassy and smooth exterior, having several stream-like cracks [6,72].

Figure 11 illustrates the SEM images of DPF/epoxy composites, revealing poor filler-matrix interface bonding along with micro-voids which explains the lower mechanical strength and modulus with respect to AA/epoxy. The AA/epoxy composites presented relatively better bonding, less fiber pull-out, fiber breakage, and delamination, compared to the A/epoxy, G/epoxy, and L/epoxy composites. Similar SEM images were observed for date palm sheath fiber-reinforced polycaprolactone biopolymer composites fabricated by the extrusion process [31]. The higher strength of the AA/epoxy composites can be explained by the perfect wettability of the AA surface in the epoxy matrix, compared to the other composites. It has already been established that unstable interfacial adhesion between fiber/filler and matrix often results in fiber pull-out, fiber-fiber delamination and matrix debonding are being more possible in the case of natural fibers than for their synthetic counterparts [31]. 

The fracture surface SEM image of A/epoxy composites shows numerous and predominant large voids, related to the higher pull-out, poor matrix wetting and delamination resulting failure mechanism, compared to the rest of the materials developed in this study. 

Comparative arguments were presented by other researchers [36]. DPF usually has a cylindrical shape, consisting of a parallel assembly of microfibrils, with considerable amounts of non-cellulosic elements on the fiber surface, including waxes, natural oils, lignin, hemicellulose, pectin, and other impurities specific to the Arabian and Saharan areas [25,26,28,31,72,73]. The cylindrical strand shape of DPF is observed in all the composites, although it is more clearly visible in the L/epoxy and G/epoxy composites, as also reported in the literature [36]. Remarkably, fibrous bundles or the honeycomb morphology are also visible in the SEM images of all the DPF/epoxy composites. Relatively similar honeycomb structures were noticed in the fracture surface morphology of unidirectional composites comprising date palm petiole fibers [32] and in the surface morphology of untreated date palm petiole fibers from the region of Tozeur (Southern Tunisia) [33]. 

## 5. Promising Applications of AA Filler/Epoxy Composites

The most promising applications of DPF/epoxy might include there usage in cost effective insulation building systems to composite roofing, wall flooring, ceilings, beams, and columns besides certain furniture and household products with high mechanical performance. It can also be extended to lightweight structural aerospace, railway, truck, sporting, and marine applications. Interior and certain exterior vehicle parts can also be made on account of better mechanical strength and minimum energy absorption tendencies such as package trays, trunk liners, seat backs, racks, spare tire linings, headliner panels, dashboards, back cushions, boot linings, windshields, noise insulation panels, and door trim panels.

## 6. Conclusions

In this study, fibers extracted from different parts of the date palm tree were used as reinforcement as filler in epoxy matrix in order to investigate their mechanical strength, density, water absorption and morphological properties. The results revealed that the incorporation of DPF, particularly AA filler, considerably improves the strength and moduli of the fabricated composite compared to the epoxy composites. Similar trends are followed in the physical properties of the developed composite. The analysis of the fractured surface morphologies showed relatively less fiber pull-out, debonding, and void content within the AA/epoxy composites, which governed their higher performance and sequentially improved properties compared to the rest of the composites. 

The findings of this study encourage the valorization of date palm biomass obtained during annual pruning, which will probably minimize waste deposition. The current study reveals the promising potential of DPF for industrial lightweight engineering and outdoor applications, including automotive parts and constructional panels. In addition, these composite materials could also be considered for energy and sound absorption applications.

## Figures and Tables

**Figure 1 materials-12-02145-f001:**
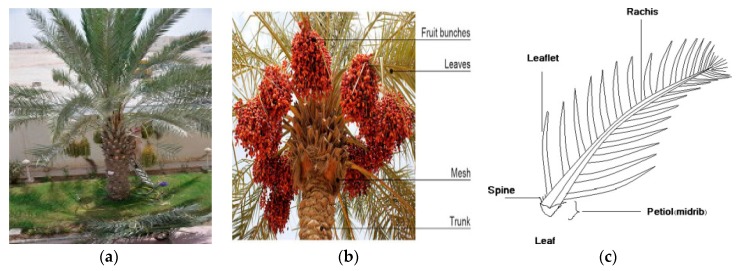
(**a**–**c**) Date pam tree [14]; (**a**) date palm tree components [15]; (**b**) and structural parts of date leaves [16] (**c**).

**Figure 2 materials-12-02145-f002:**
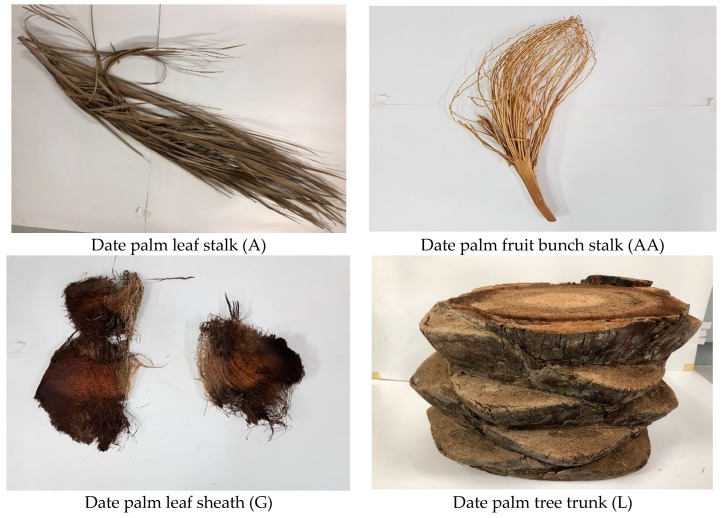
Date palm tree parts used for extracting respective fibers.

**Figure 3 materials-12-02145-f003:**
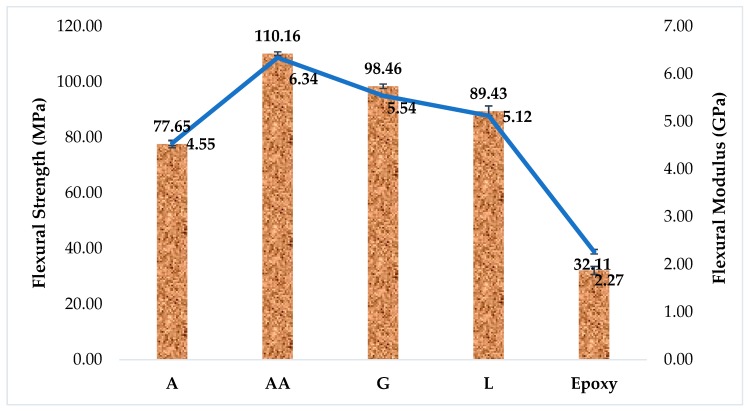
Flextural properties of DPF/epoxy composites.

**Figure 4 materials-12-02145-f004:**
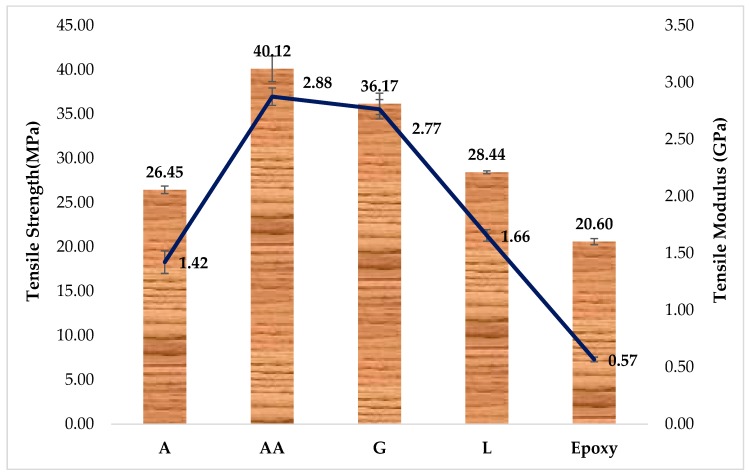
Tensile properties of DPF/epoxy composites.

**Figure 5 materials-12-02145-f005:**
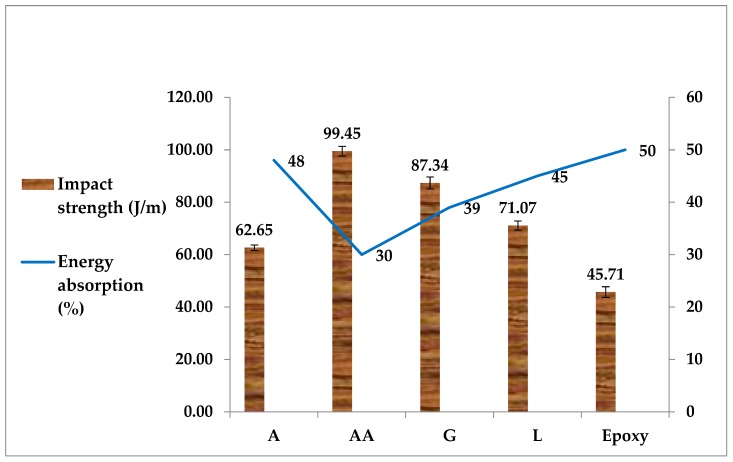
Impact strength of DPF/epoxy composites.

**Figure 6 materials-12-02145-f006:**
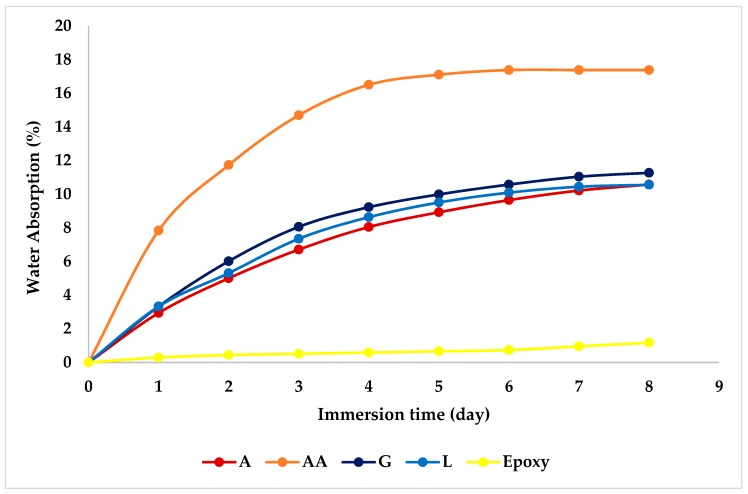
Water absorption tendencies of DPF/epoxy composites.

**Figure 7 materials-12-02145-f007:**
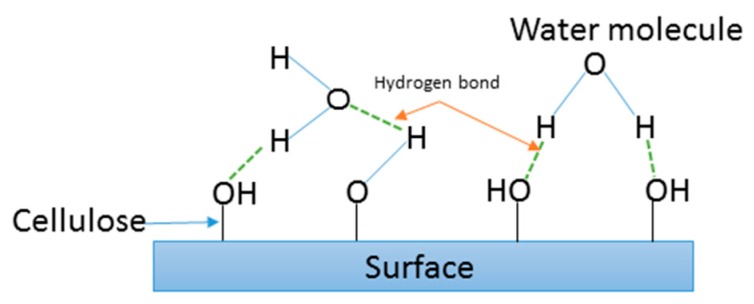
Schematic diagram for the hydrogen bond between water molecules and cellulose [24].

**Figure 8 materials-12-02145-f008:**
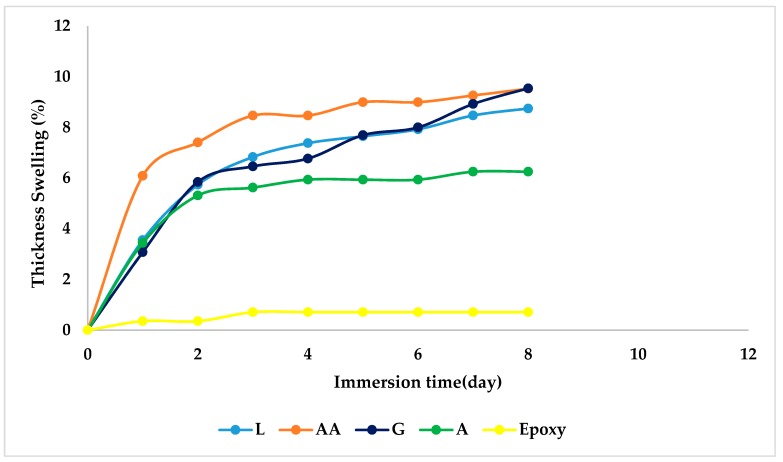
Thickness swelling of DPF/epoxy composites.

**Figure 9 materials-12-02145-f009:**
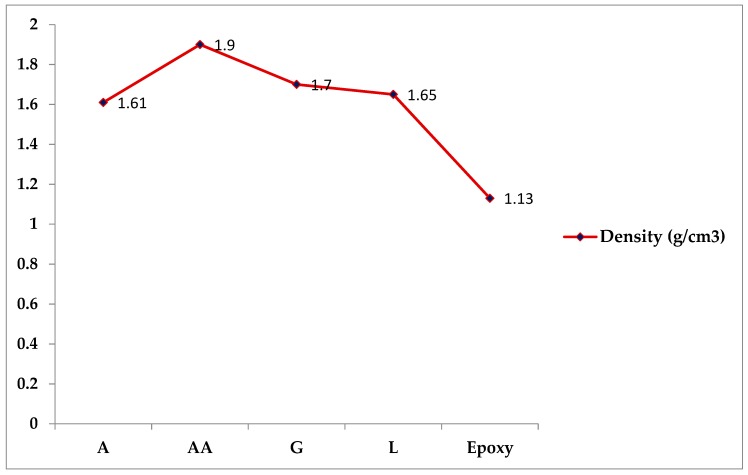
Densities of DPF/epoxy composites.

**Figure 10 materials-12-02145-f010:**
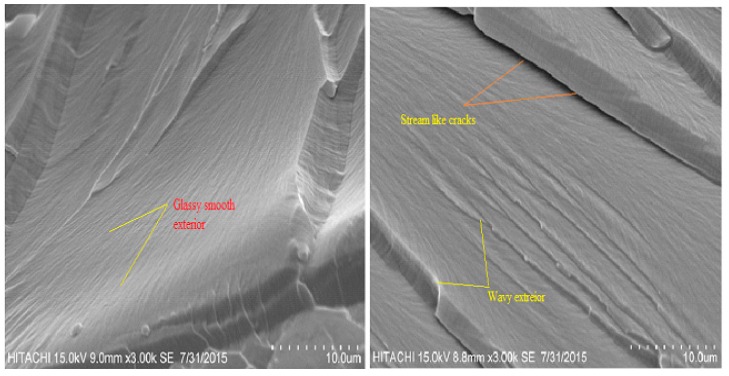
SEM images of pure epoxy composites.

**Figure 11 materials-12-02145-f011:**
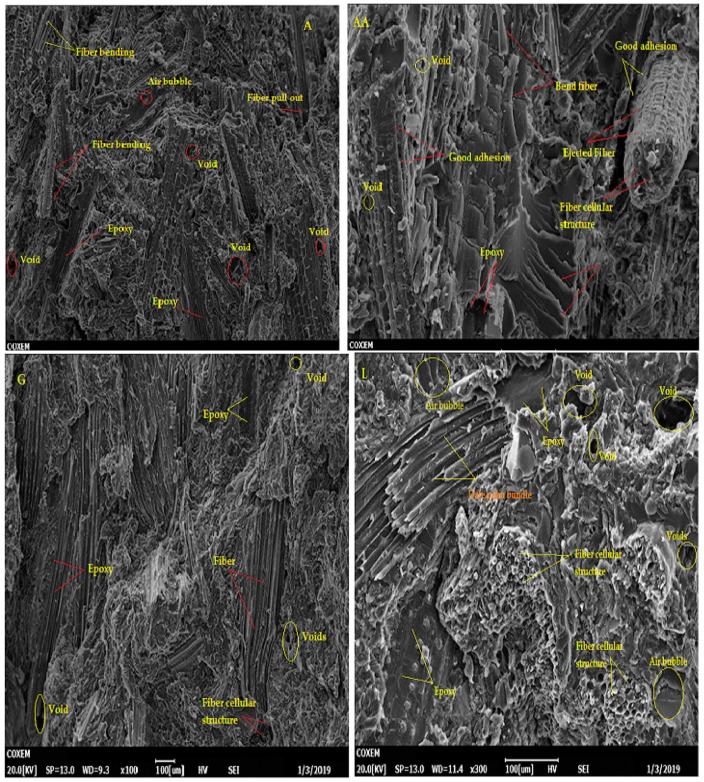
SEM images of DPF/epoxy composites.

**Table 1 materials-12-02145-t001:** Reported studies on date palm material reinforced polymer composites.

Date Palm Fibers	Polymer Matrix	References
Date palm sheath fibers	Commercial chitosan	[25]
Alkali treated DPF	Polyurethane	[26]
Alkali treated date palm leaf fibers	Recycled poly (ethylene-terephthalate)	[23]
Leaf sheath DPF	-	[27]
Date palm particles	Rigid Polyurethane	[28]
DPF (petiole, bunches and rachis)	Gypsum	[29]
Date palm spikelet	Mortar	[30]
Date palm sheath fibers	Polycaprolactone	[31]
Date palm petiole wood	Parenchyma (matrix).	[32]
Alkali treated date palm petiole fibers	-	[33]
Hybridized date palm leaf/Glass fibers	Epoxy	[34]
Date palm wastes	Linear-low density polyethylene matrix	[35]
Hybridized date palm and flax fibers	Thermoplastic starch	[36]
Pyrolysis date palm waste biochar	PP homo-polymer	[37]
Date palm mesh fibres	Cement-based mortar	[15]
Date palm stem fibres	Epoxy	[13]
Date palm branches and expanded vermiculite	Particleboard	[38]
DPF	Recycled polypropylene and LDPE/High density polyethylene ternary blends	[39]
Date palm leaflets	Expanded polystyrene	[40]
DPF/Graphite filler	Epoxy	[41]
Date palm wood flour (rachis, leaflet and leaf)	Polyethylene	[16]
Date palm wood powder	LDPE	[20]
Date palm wood powder	Recycled linear LDPE	[14]

**Table 2 materials-12-02145-t002:** Chemical composition of DPF used in this study.

DPF	Cellulose	Hemicellulose	Lignin
A (Palm tree leaf stalk)	35.00%	15.40%	20.10%
AA (Palm tree fruit bunch stalk)	44.00%	26.00%	11.00%
G (Leaf sheath fiber)	43.50%	24.00%	18.00%
L(Palm tree trunk fiber)	40.00%	9.75%	29.50%

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
