# Peer review of "Evaluation of Mechanical, Physical, and Morphological Properties of Epoxy Composites Reinforced with Different Date Palm Fillers"

_materials, 2019, doi:10.3390/ma12132145_

Round 1
Reviewer 1 Report
The introduction is well-written. Contains a lot of information and 45 citings, but many of these citings are works by authors of this publication. It demonstrates that the authors are experts in their field, but it is good practice not to cite oneself so frequently.
The work methodology is good. A few samples were prepared for the measurements, which is very important in this type of analysis. However, why was the measurement error not marked on the impact strength plot? It should be added, in my opinion.
As for water absorption measurements, only the AA sample reached a plateau. For the rest of the samples, measurements should be taken after 2 and 3 weeks, and it should be checked whether they also reach the plateau stage.
Author Response
Reviewer Comments and Authors Response
Paper title: Evaluation of Mechanical, Physical and Morphological Properties of Epoxy Composites Reinforced with Different Date Palm Fillers
Authors: Basheer A. Alshammari, N. Saba, Majed D. Alotaibi, Mohammed F. Alotibi, Mohammad Jawaid, Othman Y Alothman
The authors would like to thank the editor and the reviewers for a careful and thorough reading of this manuscript and for the thoughtful comments and constructive suggestions, which help to improve the quality of this manuscript. We have carefully addressed the reviewer suggestions and are highlighted in Blue font within the revised manuscript.
Reviewer#1
The introduction is well-written. Contains a lot of information and 45 citings, but many of these citings are works by authors of this publication. It demonstrates that the authors are experts in their field, but it is good practice not to cite oneself so frequently.
We are highly thankful to the reviewer for nice suggestions.
The work methodology is good. A few samples were prepared for the measurements, which is very important in this type of analysis. However, why was the measurement error not marked on the impact strength plot? It should be added, in my opinion.
We now added the Error bar in the Impact strength plot as per reviewer request within revised manuscript.
Sample reached a plateau. For the rest of the samples, measurements should be taken after 2 and 3 weeks, and it should be checked whether they also reach the plateau stage.
Current study does not deal the effect of water absorption on the mechanical strength of the composites. It might be the future research study.
Reviewer 2 Report
In this paper, the authors study the reinforcement effects of various natural fibers sourced from the date palm tree in polymer matrix fiber composites. The paper overall is generally well-written, with only some minor edits and formatting required. Some of the figures (especially the SEM images) should be improved, however, as they should be higher-quality. The paper overall is promising, but it needs a major revision to address some problems, as described below. Please note that all comments are meant only to help the authors and ensure the quality of the published papers in this journal and are in no way meant to be discouraging or demeaning to the authors personally.
1. The topic of this paper is interesting and useful, but I am not convinced from this manuscript that the results from this study are novel enough for a new academic journal article. This topic has been studied extensively in the past (as evidenced by the many, many previous studies referenced by the authors in their very good literature review) and much is already known about the behavior of these natural fibers. The authors need to make a stronger argument for the novelty of this work and point out some explicit new contributions to the materials science literature. The experiments seems sound and well-done, but they need to be better sold as novel contributions.
2. The authors do point out that this is a very useful plant and common in their home country, but Materials is an international journal so the paper should be written in such a way that the usefulness is clear for a wide, international audience. A good addition to the paper which may help with this is to point out that the more widespread adoption of natural fibers in re-enforced epoxy manufacturing would be more environmentally friendly and sustainable. It could help to replace some hydrocarbon usage.
3. Finally, there seems to be very little discussion of testing of the date palm fibers compared with the performance of synthetic fibers. The authors claim this as a contribution in the abstract, so it should be greatly expanded. The discussion overall is lacking (should be expanded in a revision), but this is the area most in need of attention. It would be best to provide some benchmark data, either from tests the authors did or got from the literature.
There are a few additional specific things I would suggest that the authors consider in a revision of this paper:
a. Provide the energy of the pendulum used in the IZOD tests and provide the ASTM test type uses (there are many pendulums and 4 different tests, so this needs to be specified). Also, please provide images of your samples and give a good quality image of one of the notches so the reader can judge the reliability of the IZOD test.
b. The font used in the performance plots (Figures 2-5, 7-8) should be Times New Roman or Palatino Linotype, as this is more professional. The figures also need more consistent scaling and improved presentation overall. A picture of a sample for each test should be given next to each of these figures. In addition, the authors should also add the number of replications to each of these figures to make it very clear how many tests were done.
c. The SEM images in Figure 10 are too low-quality to read easily and should be improved
d. In Figure 1 or near it, it would be helpful to have an image showing what the fibers used in the experimentals look like
e. There is some inconsistent text types throughout the document - make sure that it is consistent and flows well
f. Add a dedicated Discussion section, as there is quite a lot of interesting things to talk about with the experimental results observed
g. The Conclusions section is weak and does not really excite the reader that the work presented in this study is new and interesting - this should be improved
h. Provide official references to the standards used to drive the experiments
i. Provide a reference or two for Figure 6 so that the reader can go and find more information about this effect if they wish
j. Finally, in the discussion section, the authors should talk about potential errors from the machine resolution for the experiments. The authors state that their machine has a 100KN load cell, so the resolution is not very good for some of the tests - the Reviewer sometimes use a 2KN load cell for these kinds of experiments to improve the resolution. No additional experiments need to be done, but the authors do need to discuss and acknowledge the effect from this.
Author Response
Reviewer Comments and Authors Response
Paper title: Evaluation of Mechanical, Physical and Morphological Properties of Epoxy Composites Reinforced with Different Date Palm Fillers
Authors: Basheer A. Alshammari, N. Saba, Majed D. Alotaibi, Mohammed F. Alotibi, Mohammad Jawaid, Othman Y Alothman
The authors would like to thank the editor and the reviewers for a careful and thorough reading of this manuscript and for the thoughtful comments and constructive suggestions, which help to improve the quality of this manuscript. We have carefully addressed the reviewer suggestions and are highlighted in Blue font within the revised manuscript.
Reviewer #2
Comments and Suggestions for Authors
In this paper, the authors study the reinforcement effects of various natural fibers sourced from the date palm tree in polymer matrix fiber composites. The paper overall is generally well-written, with only some minor edits and formatting required. Some of the figures (especially the SEM images) should be improved, however, as they should be higher-quality. The paper overall is promising, but it needs a major revision to address some problems, as described below. Please note that all comments are meant only to help the authors and ensure the quality of the published papers in this journal and are in no way meant to be discouraging or demeaning to the authors personally.
All authors are highly thankful for nice support and motivational comments.
The topic of this paper is interesting and useful, but I am not convinced from this manuscript that the results from this study are novel enough for a new academic journal article. This topic has been studied extensively in the past (as evidenced by the many, many previous studies referenced by the authors in their very good literature review) and much is already known about the behavior of these natural fibers. The authors need to make a stronger argument for the novelty of this work and point out some explicit new contributions to the materials science literature. The experiments seem sound and well-done, but they need to be better sold as novel contributions.
The novelty of this study is that it deals with local materials to fabricate composites by using different parts of Date Palm tree to investigate their performance as reinforcements for polymer matrix.
The authors do point out that this is a very useful plant and common in their home country, but Materials is an international journal so the paper should be written in such a way that the usefulness is clear for a wide, international audience. A good addition to the paper which may help with this is to point out that the more widespread adoption of natural fibers in re-enforced epoxy manufacturing would be more environmentally friendly and sustainable. It could help to replace some hydrocarbon usage.
We added the suggested point to the Last paragraph of Introduction section within revised manuscript as per reviewer comment.
Finally, there seems to be very little discussion of testing of the date palm fibers compared with the performance of synthetic fibers. The authors claim this as a contribution in the abstract, so it should be greatly expanded.
As this article deals about the performance of different date palm fibers, so we now rectified and delete the word Synthetic fibers from the Abstract within the revised manuscript as per reviewer suggestions.
The discussion overall is lacking (should be expanded in a revision), but this is the area most in need of attention. It would be best to provide some benchmark data, either from tests the authors did or got from the literature.
There are a few additional specific things I would suggest that the authors consider in a revision of this paper:
Provide the energy of the pendulum used in the IZOD tests and provide the ASTM test type uses (there are many pendulums and 4 different tests, so this needs to be specified). Also, please provide images of your samples and give a good quality image of one of the notches so the reader can judge the reliability of the IZOD test.
We have now introduced the required detail regarding the IZOD notch test, within the revised manuscript as per reviewer comment.
The font used in the performance plots (Figures 2-5, 7-8) should be Times New Roman or Palatino Linotype, as this is more professional. The figures also need more consistent scaling and improved presentation overall. A picture of a sample for each test should be given next to each of these figures. In addition, the authors should also add the number of replications to each of these figures to make it very clear how many tests were done.
We have now changed Font used in the performance plots of all Figures 2-9 in Palatino Linotype and tried to improve the presentation overall as well, according to reviewer request.
The SEM images in Figure 10 are too low-quality to read easily and should be improved
We have now added high Quality SEM images of Figure 10 however it become Figure 11 in the revised manuscript, according to reviewer request.
In Figure 1 or near it, it would be helpful to have an image showing what the fibers used in the experimental look like
As after extraction they look alike hence it is quite hard to find out difference from our naked eye.
So we have added the Date palm tree parts fiber Source from where we extract the fibers, within the revised manuscript in Figure 2 to shows the difference.
There is some inconsistent text types throughout the document - make sure that it is consistent and flows well
We have gone through the whole manuscript to make the flow consistent within the revised manuscript.
f. Add a dedicated Discussion section, as there is quite a lot of interesting things to talk about with the experimental results observed
We have now improved the Discussion Section as per reviewer comments.
g. The Conclusions section is weak and does not really excite the reader that the work presented in this study is new and interesting - this should be improved.
We have now rewritten the Conclusion section as per reviewer request.
h. Provide official references to the standards used to drive the experiments
We have now added an official references to the ASTM standards for the tests used in this study.
i. Provide a reference or two for Figure 6 so that the reader can go and find more information about this effect if they wish
We have now added reference for Figure 6 however it becomes Figure 7 within the revised manuscript.
j. Finally, in the discussion section, the authors should talk about potential errors from the machine resolution for the experiments. The authors state that their machine has a 100KN load cell, so the resolution is not very good for some of the tests - the Reviewer sometimes use a 2KN load cell for these kinds of experiments to improve the resolution. No additional experiments need to be done, but the authors do need to discuss and acknowledge the effect from this.
We have now rectified the Machine load cell used in this test to avoid any confusion within the revised manuscript according to reviewer requests.
Reviewer 3 Report
This work shows interesting results in the evaluation of epoxy composites with date palm fibers.
It could be acceptable for publication but need some revisions.
Indeed, some precisions could be added in experimental procedure. Also, to support the statement that mechanical properties decreased after water sorption, more analysis could be done, e.g. before-and-after comparison. The font-size should be uniformized in all the text.
See underline text with comments in joined pdf file.

Author Response
Reviewer Comments and Authors Response
Paper title: Evaluation of Mechanical, Physical and Morphological Properties of Epoxy Composites Reinforced with Different Date Palm Fillers
Authors: Basheer A. Alshammari, N. Saba, Majed D. Alotaibi, Mohammed F. Alotibi, Mohammad Jawaid, Othman Y Alothman
The authors would like to thank the editor and the reviewers for a careful and thorough reading of this manuscript and for the thoughtful comments and constructive suggestions, which help to improve the quality of this manuscript. We have carefully addressed the reviewer suggestions and are highlighted in Blue font within the revised manuscript.
Reviewer#3
Comments and Suggestions for Authors
This work shows interesting results in the evaluation of epoxy composites with date palm fibers. It could be acceptable for publication but need some revisions.
All the authors are highly thankful to the reviewer for their nice wordings.
Indeed, some precisions could be added in experimental procedure. Also, to support the statement that mechanical properties decreased after water sorption, more analysis could be done, e.g. before-and-after comparison. The font-size should be uniformized in all the text.
We have made more analysis as per reviewer comments.
We have now uniformized the font size throughout the text as per reviewer suggestions.
See underline text with comments in joined pdf file.-peer-review-4398506.v1.pdf
We have gone through all the comments in the joined pdf-file provided by the reviewer and corrected it within the revised manuscript.
Distance between plates? Mold Size
We have now added the information regarding mold size within the revised manuscript.
You should discuss about energy absorption... otherwise it is useless to show the results in Figure 4.
We have now added the discussion about energy absorption as per reviewer request.
It could be interesting to test the mechanical properties of the different composites after water absorption test to verify this statement?
Current study deals about the comparative mechanical properties of different DPF in epoxy resin. It does not deal the effect of water absorption on mechanical properties of composites; it would be a future study in this research.
It is not easy to see each zone. You should show the different parts more distinctly, maybe with different colors? If possible, could you provide pictures with better resolution?
We now added the improved SEM pictures epoxy composites (Figure 10) as per reviewer request within the revised manuscript.
Round 2
Reviewer 2 Report
The authors have done a very good job overall with the revision of this paper. It is significantly improved. However, I would like to point out that there are still a few issues that should be addressed before it would be acceptable for publication. Please see below. The paper should be sent back to the authors for a few more revisions (minor revision):
1. It is generally bad practice to use first- or second-person voice in technical papers (for example, don't say "I" or "we" or "us" - just state the facts for the reader). Please consider addressing this. It is not grounds for rejecting a paper but it is much more professional to use third-person voice only (i.e., say "the results showed X" instead of "our work showed X").
2. Thank you for updating the information about the IZOD tests. However, I must point out that a 10 KJ pendulum is meant for ductile metals like aluminum and copper with very high fracture toughness. It is definitely not an appropriate testing apparatus for polymer materials in most cases. The IZOD machine that I use for my research into polymer materials is just 4J (some of my materials have impact strength up to 500 J/m so it would certainly work for your materials) - a 10KJ pendulum just has terrible resolution for such low impact strength materials. Your results make sense and you seem to have collected useful data, so you do not need to remove this section from the paper, but it should be very clearly pointed out that the test and data has limitations and explain the machine. If the authors plan to continue polymer testing with IZOD, they should consider purchasing or building a much smaller pendulum to use for work like this (I suggest 4 or 5 J).
3. Please state which ASTM IZOD test you did - there is Type A, Type B, Type C, and Type E. It is very important to know this in order to evaluate your results. I am making a guess from your data that it is Type A, but it needs to be stated. Type A and Type E are the most common and the major difference between them is that Type A opens the notch (tension) on striking and Type E closes it (compression). Type E is commonly used for anisotropic materials and studying material boundaries (for example, material that has been surface heat-treated or 3-D printed materials).
4. Please include a microscope image of one of the notches for your IZOD samples. The notch geometry is the most important factor for the performance of this test.
5. Figures 6, 8, and 9 are grainy and should be improved with higher-resolution images
6. In a few places in a paper, the authors state that a figure or dataset "clearly" show something. I tend to agree in the cases I observed, but be careful that you really want to make that statement as you may be challenged on the statement if it isn't completely true. Double check in the final version that you want to make a claim this strong for each of the use cases.
Great job with the revision and good luck with your continued research!
Author Response
Editor-Reviewer Comments and Author Response
Paper Title: Evaluation of Mechanical, Physical and Morphological Properties of Epoxy Composites Reinforced with Different Date Palm Fillers
Authors: Basheer A. Alshammari, N. Saba, Majed D. Alotaibi, Mohammed F. Alotibi, Mohammad Jawaid, Othman Y Alothman
The authors would like to thank the editor and the reviewers for a careful and thorough reading of this manuscript and for the thoughtful comments and constructive suggestions, which help to improve the quality of this manuscript. We have carefully addressed the Editor suggestions and are highlighted in Blue font within the revised manuscript.
Reviewer# 2
The authors have done a very good job overall with the revision of this paper. It is significantly improved.
All authors are highly thankful for nice appreciation!
However, I would like to point out that there are still a few issues that should be addressed before it would be acceptable for publication. Please see below. The paper should be sent back to the authors for a few more revisions (minor revision):
It is generally bad practice to use first- or second-person voice in technical papers (for example, don't say "I" or "we" or "us" - just state the facts for the reader). Please consider addressing this. It is not grounds for rejecting a paper but it is much more professional to use third-person voice only (i.e., say "the results showed X" instead of "our work showed X").
Authors have now revised the manuscript according to reviewer comments!
Thank you for updating the information about the IZOD tests. However, I must point out that a 10 KJ pendulum is meant for ductile metals like aluminum and copper with very high fracture toughness. It is definitely not an appropriate testing apparatus for polymer materials in most cases. The IZOD machine that I use for my research into polymer materials is just 4J (some of my materials have impact strength up to 500 J/m so it would certainly work for your materials) - a 10KJ pendulum just has terrible resolution for such low impact strength materials. Your results make sense and you seem to have collected useful data, so you do not need to remove this section from the paper, but it should be very clearly pointed out that the test and data has limitations and explain the machine. If the authors plan to continue polymer testing with IZOD, they should consider purchasing or building a much smaller pendulum to use for work like this (I suggest 4 or 5 J).
We realized that Pendulum used in this testing is 5 J not 10 KJ. We have now rectified this error which occurs during revision.
Please state which ASTM IZOD test you did - there is Type A, Type B, Type C, and Type E. It is very important to know this in order to evaluate your results. I am making a guess from your data that it is Type A, but it needs to be stated. Type A and Type E are the most common and the major difference between them is that Type A opens the notch (tension) on striking and Type E closes it (compression). Type E is commonly used for anisotropic materials and studying material boundaries (for example, material that has been surface heat-treated or 3-D printed materials).
We used Type A Izod Test in this work. We have now added the information regarding this issue within revised manuscript according to your suggestion.
Please include a microscope image of one of the notches for your IZOD samples. The notch geometry is the most important factor for the performance of this test.
We have added the information regarding the notch geometry, but unfortunately we have not taken microscope image of the requested IZOD samples and its not possible to do at this moment.
Figures 6, 8, and 9 are grainy and should be improved with higher-resolution images
We have now added higher resolution images of Figure 6, 8, 9 within the revised manuscript.
In a few places in a paper, the authors state that a figure or dataset "clearly" show something. I tend to agree in the cases I observed, but be careful that you really want to make that statement as you may be challenged on the statement if it isn't completely true. Double check in the final version that you want to make a claim this strong for each of the use cases.
We have thoroughly revised the manuscript according to reviewer suggestion!
Great job with the revision and good luck with your continued research!
All authors are thankful to the reviewer for giving constructive suggestion to improve the manuscript.